# View of Times and Temporal Focus under the Pace of Life on the Impact of Intertemporal Decision Making

**DOI:** 10.3390/ijerph20054301

**Published:** 2023-02-28

**Authors:** Yue Wang, Xiaoyu Wang, Xiao Yang, Fangyuan Yuan, Ying Li

**Affiliations:** School of Education, Zhengzhou University, Zhengzhou 450001, China

**Keywords:** pace of life, view of time, temporal focus, intertemporal decision making

## Abstract

Previous studies have explored the effects of time poverty and money worship on intertemporal decision making based on a resource scarcity perspective. However, how the pace of life affects intertemporal decision making has not been examined. Furthermore, manipulating time perceptions can influence intertemporal decision-making preferences. Based on the perspective of time perception differences, it remains unknown how views of time or temporal focus affect the intertemporal decision making of individuals with different pace of life. To address these issues, study 1 adopted a correlational study to initially explore the relationship between the pace of life and intertemporal decision making. Studies 2 and 3 used manipulation experiments to examine the effects of the pace of life and view of time and temporal focus and pace of life on intertemporal decision making. The results suggest that the faster the life pace, the more recent rewards are preferred. Views of time and temporal focus manipulations can influence the intertemporal decision making of faster-paced individuals, making them prefer smaller–sooner (SS) payoffs under a linear view of time or future temporal focus and larger–later (LL) payoffs under a circular view of time or past temporal focus. However, the manipulation does not affect the intertemporal decision of slower-paced individuals. Our study examined the effect of the pace of life on intertemporal decision making based on a resource scarcity perspective, and found boundary conditions for the influence of the view of time and temporal focus on intertemporal decision making based on the perspective of differences in people’s perception of time.

## 1. Introduction

Intertemporal decision making refers to decision behavior when weighing the gains and losses that occur at different time points [1]. It is crucial for people’s rational consumption and healthy life [2] and even the conservation of clean air to improve public health [3]. Intertemporal decision making refers to decisions involving outcomes available at different points in time, usually between a smaller/sooner (SS) reward and a larger/later (LL) reward [4,5]. Usually, people manifest a strong preference for immediate outcomes, and future outcomes are devalued as a function of delay, a process that is referred to as delay discounting [6]. Previous studies have explored the effects of time poverty [7], death awareness [8], and money scarcity [9] on intertemporal decision making from a resource scarcity perspective. This suggests that scarcity of time or money resources can influence intertemporal decision making.

As the saying goes “Time is money and efficiency is life” [10]. With the development of society, peoples’ lives are becoming more and more fast-paced, and individuals perceive that time is becoming insufficient and time scarcity is becoming more and more serious [7,11], which will lead to a series of psychology changes. Studies have shown that the scarcity of time resources can weaken one’s self-control, and the lack of time creates a poverty mindset, which leads to a poverty mindset that prompts individuals to unconsciously focus on the task at hand [7]. For individuals with a fast pace of life, they may not be able to delay gratification due to their weak self-control abilities and focus on the current task. On the contrary, for individuals with a slow pace of life, relatively adequate time resources and greater self-control may make them more willing to delay gratification. However, to our knowledge, no studies have directly examined the relationship between the pace of life and intertemporal decision making to date. Study 1 adopted a correlational study to initially explore the relationship between the pace of life and intertemporal decision making. Studies 2 and 3 used an experimental approach to further explore the causal relationship between the pace of life and intertemporal decision making. Moreover, hypothesis 1 is proposed based on the above discussion: the faster the pace of life, the more likely they were to prefer small near-term rewards, and the slower the pace of life, the more likely they were to prefer large far-term rewards (Hypothesis 1).

Individuals’ intertemporal decision preferences are not set in stone. The fact that manipulating the way they perceive time can change individuals’ intertemporal decision-making preference has been extensively studied [12,13]. Individuals’ view of time can be classified into two types, namely, linear and circular views of time. Individuals who hold a linear view of time believe that time will never return and tend to regard time as a linear one-way movement that constantly extends forward, develops and changes, and will pass. Individuals with a circular view of time believe that things change periodically over time and tend to understand time as a circular motion that keeps repeating [14,15,16]. By manipulating participants’ view of time, Xu et al. (2019) found that people with a linear view of time expected greater future changes, had longer participant time judgments on the same time interval, had greater time discount rates, and preferred SS. The opposite was true for individuals with a circular view of time [12]. As it is known that individuals with a fast pace of life are more reluctant to delay gratification than those with a slow pace of life, therefore, can the manipulation of the view of time change the intertemporal decision preference for individuals with different paces of life? Thus, we can guide individuals with different paces of life to make rational decisions. We conjecture that the pace of life and view of time can jointly influence intertemporal decision making, with fast-pace-of-life and linear-view-of-time individuals preferring SS and slow-pace-of-life, circular-view-of-time individuals preferring LL (Hypothesis 2). Study 2 used an experimental approach to test Hypothesis 2.

Temporal focus is the extent to which individuals characteristically direct their attention to the past, present, and/or future [17], which is a cognitive attitude towards time itself. Scholars have also found that temporal focus can influence individuals’ resource allocation. For example, Nguyen et al. (2021) showed that temporal focus can influence the philanthropic commitment of CEOs of state-owned and privately owned firms. Specifically, CEOs of state-owned firms have less philanthropic commitment when they hold a future-oriented focus, while the exact opposite is true for CEOs of privately owned firms [18]. Subsequently, by manipulating individuals’ temporal focus to make them focus on the past or the future, the researchers found that individuals who focused on the future (future focus) preferred small near-term rewards more in intertemporal decision making, while individuals who formed a past temporal focus preferred large far-term rewards [19]. In addition, the faster the pace of life, the more they may focus on the present (present-focused) [7]; in other words, they may hold a temporal focus with a preference for timely gratification. As previously mentioned, temporal focus can influence decision preferences. In consequence, the third question we pay attention to is whether temporal focus manipulation can affect the intertemporal decision preferences of individuals with different paces of life, and more specifically, how the pace of life and temporal focus interact with intertemporal decision making. We argue that the pace of life and temporal focus can jointly influence intertemporal decision making and a fast-pace-of-life, future-focus-preference SS, as well as a slow-pace-of-life, past-focus preference LL. (Hypothesis 3). Study 3 used an experimental approach to test Hypothesis 3.

Summing up, after a review of some critical papers on intertemporal choices, we argue for the necessity of addressing three issues that need further elaboration in the current delay-discounting literature: (1) the effect of the pace of life on intertemporal decision making, (2) the influence of the view of time manipulations on the intertemporal decision making preferences of individuals with different paces of life and (3) the influence of temporal focus manipulations on the intertemporal decision making preferences of individuals with a different pace of life. Study 1 firstly explored the relationship between the pace of life and intertemporal decision making using a correlational study. Studies 2 and 3 explored how the pace of life and individual time perception difference factors (view of time or temporal focus) interacted to influence intertemporal decision making by changing the view of time or temporal focus of individuals with different paces of life through a manipulation experiment. The following are hypotheticals.

**Hypothesis** **1.**
*the faster the pace of life, the more likely they were to prefer small near-term rewards, and the slower the pace of life, the more likely they were to prefer large far-term rewards.*


**Hypothesis** **2.**
*pace of life and time of view can jointly influence intertemporal decision making, with fast-pace-of-life. Linear-view-of-time individuals preferring SS and slow-pace-of-life, circular-view-of-time individuals preferring LL.*


**Hypothesis** **3.**
*pace of life and temporal focus can jointly influence intertemporal decision making and fast-pace-of-life, future-focus preference SS, as well as slow-pace-of-life, past-focus preference LL.*


Based on previous research, our study examines the effects of fast and slow paces of life on intertemporal decision making, enriching the study of factors influencing intertemporal decision making. In addition, we examine how the manipulation of time perception factors changes the intertemporal decision making of individuals with different paces of life. This provides a basis for individuals with different paces of life to cope with changes in life, adjust themselves, and make appropriate intertemporal decisions. Moreover, by focusing on the role of time perception manipulation on intertemporal decision making, our study could constitute a building block for successful future manipulation programs targeted at mental and physical health issues, including gambling behavior [20].

## 2. Study 1: A Preliminary Investigation of the Relationship between Pace of Life and Intertemporal Decision Making

Study 1 initially explored the correlation between the pace of life and intertemporal decision making using the classic intertemporal decision-making task.

### 2.1. Participants

The questionnaire was administered to 991 students from freshman to graduate school in three universities in Henan Province using a convenience sampling method. Finally, 799 (501 females and 298 males, *M*_age_ = 20.79 ± 4.44 years) valid questionnaires were collected, with effective recovery rate of 80.63%.

### 2.2. Materials and Methods

#### 2.2.1. Materials 

Pace of Life Questionnaire developed by Wiseman (2006) was used to examine the participants’ pace of life [21]. The questionnaire contains 7 questions (e.g., “Are you the first person to finish at mealtimes?”). The options include “Often (10 points)”, “Sometimes (5 points)”, and “Never (1 point)”. The higher the score, the faster the pace of life. The Chinese version of the questionnaire was translated by several graduate students in psychology (including those who passed the fourth and eighth grade of English), back-translated by a foreign language faculty member, and then finalized by the group after repeated discussions several times. Cronbach α= 0.60 for this study.

The intertemporal decision-making questionnaire was developed by Chen and He (2011) [22] and it contains 19 questions. The questionnaire asks participants to choose between a smaller (SS) and a longer-term (LL) reward, either today or six months from now, with SS as option A and LL as option B. SS(A) starts at ¥50 and increases by ¥50 each time to ¥950, while LL(B) remains constant at ¥1000. The participant value of “¥1000 after 6 months” (the delayed option) was taken as the average of the amount of option A when the participant first chose A and the amount of option A in the previous question.

#### 2.2.2. Procedure

Study 1 used the Questionnaire Star platform to collect data online. Participants were from two universities in Henan Province. During the survey process, participants filled in demographic information first, followed by the Pace of Life Scale and the Intertemporal Decision-making Questionnaire in that order. Afterwards, we removed the data from the questionnaires that did not pass the polygraph questions. The average time to complete the questionnaire was 167.33 s.

### 2.3. Results

SPSS 21.0 (IBM, Armonk, NY, USA) was used for data collation and analysis. Calculate the time discount rate K using the hyperbolic discount model V = A/(1 + KD) (V = participant value, A = delay amount, D = delay time (days)). A logarithmic transformation of K yielded lnK, with larger lnK values giving participants a preference for smaller recent rewards. The results showed that *M*_pace of life_ = 30.86 (*SD* = 8.95), *M*_time discount rate_ = −5.68 (*SD* = 2.27), and the individual’s pace of life score was significantly and positively correlated with the time discount rate (*r* = 0.141, *p* = 0.000 < 0.001). The faster the individual’s pace of life, the higher his or her participant delay discount rate and the more inclined he or she was to choose small rewards in the near future. The slower the individual’s pace of life, the lower their participant delay discount rate and the more inclined they were to choose large rewards in the distant future. This result confirms Hypothesis 1.

## 3. Study 2: The Influence of the Pace of Life and View of Time on Individual Intertemporal Decision Making

Study 1 has found a correlation between the pace of life and intertemporal decision making. Study 2 used an experimental approach to further explore the effect of the pace of life on intertemporal decision making and how the pace of life and view of time affect individuals’ intertemporal decision making by manipulating in the linear or circular view of time of individuals with different paces of life.

### 3.1. Materials and Methods

#### 3.1.1. Participants

According to G*Power 3.1 at significance level α = 0.05 and medium effect size (*f* = 0.25), the total sample size to achieve 80% statistical test power is at least 128. The sample size for this experiment met the criteria. Undergraduate and graduate students were recruited at three universities with a total of 303 participants (145 females and 158 males, *M*_age_ = 19.83 ± 2.83 years). The mean score of participants on the pace of life was calculated according to the pace of life scale as 30.34, and a score greater than 31 was defined as the faster pace of life group and less than 30 as the slower pace of life group. An independent samples *t*-test was used to show that the scores of the faster-pace-of-life participants (*M* = 38.80, *SD* = 6.35) were significantly higher than those of the slower-pace-of-life participants (*M* = 21.94, *SD* = 5.84), *t* (301) = −24.06, *p* = 0.000, Cohen’s d = 2.76, group validity. Table 1 shows the grouping information of the participants. The participants were paid after the experiment. The study was approved by the local Ethics Committee of Zhengzhou University. 

#### 3.1.2. Design

This study adopted a 2 (Pace of life: faster vs. slower) × 2 (View of time: linear vs. circular) between-participant design. The dependent variable was the log-transformation of the time discount rate (lnK), with larger values of lnK indicating participants’ greater preference for SS.

#### 3.1.3. Materials

Time-view manipulation tasks. Referring to Xu et al. (2019), the participants’ different views of time were manipulated by completing an utterance manipulation task and an event recall task [12]. In the linear view of time task, for example, the participants were asked to read a known poem and select the appropriate option to fill in (e.g., “The years are like a shuttle, time is like an arrow”). After that, participants were asked to read the relevant material and then perform an event recall task. Material 1 of the linear view of time was as follows: “Our daily life is made up of individual periods, just as we leave childhood, enter adolescence, and move from adolescence to adulthood. It is impossible to start our lives over again, and many things disappear once they have passed and do not repeat themselves”. Resource 2 asked participants to read a picture (see Figure 1) and a text description related to a linear view of time, as follows: “The picture above shows the growth of a fruit tree, moving forward from buds, to growing taller, to dense leaves, and then to fruit. Our life is also like the tree, from birth to infancy, then from childhood to adolescence, and slowly into adulthood. In this process, our time keeps flowing forward, and we cannot stop or go back to the past.“ After reading the above information, the participants were asked to recall instances in their lives that corresponded to it and record them. The manipulation task for the circular-time-view group was similar to that of the linear-time-view group, except that it differed in the manipulation of pictures and the description of statements.

The view of time questionnaire was used to test the validity of the manipulation of participants’ different views of time. The questionnaire was adapted by Xu et al., with Cronbach’s α = 0.89 [8]. The questionnaire contains six questions, the first three questions reflecting a linear view of time (e.g., I feel that time is constantly passing.), and the last 3 questions reflecting a circular view of time (e.g., I feel that time can go around and around.) [12]. A 7-point Likert scale was used for scoring.

The pace of life questionnaire and the intertemporal decision-making task were the same as in Study 1.

#### 3.1.4. Procedure

To ensure validity of the experiment, we concealed the intention of the experiment from the participants. Demographic information on the participants was collected prior to the start of the experiment. In the experiment, participants sequentially completed (1) the pace of life questionnaire, (2) two time-view manipulation tasks and the utterance manipulation and event recall tasks (manipulating participants’ time views), (3) the time-view questionnaire (testing the validity of the manipulation), and (4) the intertemporal decision-making task questionnaire.

### 3.2. Results

SPSS 21.0 was used for data collation and analysis.

#### 3.2.1. Manipulation Check

The validity of the temporal perspective manipulation was examined using the Temporal Perspective Questionnaire. The scores of the last three questions were reverse scored and averaged with the scores of the first three questions to represent the participants’ time perspective tendency scores. Higher scores represent participants’ tendency to view time in a linear way, while lower scores represent participants’ tendency to view time in a circular way [8]. The results showed that the scores of participants in different time perspective groups differed significantly, *M*_linear_ = 5.08 ± 0.64; *M*_circular_ = 3.86 ± 0.60, *t* (1301) = 17.17, *p* = 0.000, Cohen’s d = 1.97, and the manipulation was valid.

#### 3.2.2. Main Effects Analysis

Two-way ANOVA was conducted with the type of pace of life and the type of view of time as independent variables and the log-transformation of the time discount rate (lnK) as the dependent variable. The results showed a significant main effect of pace of life type *F* (1, 299) = 59.31, *p* = 0.000, $\eta_{p}^{2}$ = 0.17. The faster the pace of life, the more participants preferred SS. There was a significant main effect of view of time type *F* (1, 299) = 35.11, *p* = 0.000, $\eta_{p}^{2}$ = 0.11, meaning that linear view of time participants preferred SS over circular time-view participants.

#### 3.2.3. Interaction Analysis

A 2 (Pace of life: faster vs. slower) × 2 (View of time: linear vs. circular) two-way ANOVA showed significant two-factor interaction, *F* (1, 299) = 32.37, *p* = 0.000, $\eta_{p}^{2}$ = 0.10. The results of the simple effects analysis indicated that when participants were living at a faster pace, the linear time perspective group (*M* = −4.49, *SD* = 1.74) preferred larger gains in the far future than the circular time perspective group (*M* = −6.47, *SD* = 1.36), *t* (1, 299) = 65.46, *p* = 0.000. However, when participants were living at a slower pace, the linear time perspective group (*M* = −6.77, *SD* = 1.46) and the circular time perspective group (*M* = −6.81, *SD* = 1.35) were not significantly different, *t* (1, 299) = 0.03, *p* = 0.87. Results are shown in Table 2 and Figure 2.

The results of Study 2 also found that the pace of life can influence intertemporal decision making. More specifically, the view of time can influence intertemporal decision making, and individuals with a linear view of time prefer SS over those with a circular view of time. Both the pace of life and view of time have an impact on individuals’ intertemporal decision-making preferences. Specifically, individuals with a faster pace of life and linear view of time preferred SS, and individuals with a slower pace of life and circular view of time preferred LL. However, individuals with slower pace of life were less influenced by their view of time, and there was no significant difference in their intertemporal decision preferences regardless of whether they initiated a linear or circular view of time. Hypothesis 2 was confirmed.

## 4. Study 3: The Effects of the Pace of Life and Temporal Focus on Individual Intertemporal Decision Making

Study 2 found that the view of time manipulation changed intertemporal decision preferences for faster-paced individuals, but not for slower-paced individuals. The experimental logic of Study 3 was the same as that of Study 2. We used the classical intertemporal decision-making task to examine how the pace of life and temporal focus interact to influence intertemporal decision making by manipulating the past or future temporal focus of individuals with different paces of life.

### 4.1. Materials and Methods

#### 4.1.1. Participants

According to G*Power 3.1 at significance level α = 0.05 and medium effect size (*f* = 0.25), the total sample size to achieve 80% statistical test power is at least 128. The sample size for this experiment met the criteria. Undergraduate and graduate students were recruited at three universities, with a total of 274 participants (181 females and 93 males, *M*_age_ = 20.01 ± 3.07 years). The mean score of participants on the pace of life was calculated according to the pace-of-life scale as 30.32, and a score greater than 31 was defined as the faster-pace-of-life group and less than 30 as the slower-pace-of-life group. An independent samples *t*-test was used to show that the scores of faster-pace-of-life participants (*M* = 37.64, *SD* = 6.02) were significantly higher than those of slower-pace-of-life participants (*M* = 23.11, *SD* = 4.67), *t* (272) = −22.34, *p* = 0.000, Cohen’s d = 2.88, group validity. Table 3 shows the grouping information of the participants. The participants were paid after the experiment. The study was approved by the local Ethics Committee of Zhengzhou University.

#### 4.1.2. Design

This study adopted a 2 (Pace of life: faster vs. slower) × 2 (Temporal focus: past vs. future) between-participant design. The dependent variable was the log-transformation of the time discount rate (lnK), with larger values of lnK indicating participants’ greater preference for SS.

#### 4.1.3. Materials

Time-focused manipulation tasks were used to manipulate participants through the utterances [23]. Each of the writing exercises consisted of 10 questions. The past-focus exercise prompted participants to write about their past (e.g., “Were you happy as a child?”), and the future-focus exercise prompted them to write about their future (e.g., “Do you think you will be happy as an old person?”) The participants were asked to imagine or recall different situations described in the questions and to write down the answers to the questions in as much detail as possible.

The Chinese version of the Temporal Focus Scale (TFS), compiled by Shipp et al. 2009 and adapted by Liu and Zhang (2016) [24,25], was used to investigate the participants’ temporal focus preferences. After examining, this scale has proved to possess good construct validity and with Cronbach’s α = 0.92. The scale contains a total of 12 questions, describing the past, present, and future respectively. Among them, eight questions describing the future (e.g., “I have a vision of my future”) and past (e.g., “I remember my childhood”) temporal focus are selected in this study. The temporal-focus index (TFI) is calculated using a seven-point scale (from 1 = never to 7 = always). TFI = (Mean Future Focus Items–Mean Past Focus Items)/(Mean Future Focus Items + Mean Past Focus Items). The TFI was used to index each participant’s overall agreement with past- and future-focus-related topics on a scale from −1 (strong past focus) to +1 (strong future focus).

The pace of life questionnaire and the intertemporal decision-making task were the same as in Study 1.

#### 4.1.4. Procedure

To ensure validity of the experiment, we concealed the intention of the experiment from the participants. Demographic information on the participants was collected prior to the start of the experiment. In the experiment, participants were asked to successively complete the pace of life scale, the temporal focus manipulation task (manipulating participants with different temporal focuses), the temporal focus scale (testing the validity of manipulation with different temporal focuses), and the experimental questionnaire for the intertemporal decision-making task.

### 4.2. Results

SPSS 21.0 was used for data collation and analysis.

#### 4.2.1. Manipulation Check

The validity of temporal focus manipulation was examined using the temporal focus scale. According to the results of the temporal focus index, *M*_past_ = −0.02 ± 0.13, *M*_future_ = 0.10 ± 0.08, *t* (272) = 9.12, *p* = 0.000, Cohen’s d = 1.11, it successfully initiated the participants’ different temporal focus.

#### 4.2.2. Main Effects Analysis

Two-way ANOVA was performed on the sample using the type of pace of life and type of time focus as independent variables and the log-transformation of the time discount rate (lnK) as the dependent variable. The results showed a significant main effect of the pace of life type, *F* (1, 270) = 9.76, *p* = 0.002, $\eta_{p}^{2}$ = 0.04. Faster-pace-of-life participants (*M* = −5.13 ± 1.80) preferred SS more than slowerpace -of-life participants (*M* = −5.67 ± 1.38). A significant main effect of the time focus type *F* (1, 270) = 22.99, *p* = 0.000, $\eta_{p}^{2}$ =0.08. Past temporal focus participants (*M* = −5.82 ± 1.27) preferred LL over future temporal focus participants (*M* = −4.98 ± 1.82).

#### 4.2.3. Interaction Analysis

A 2 (Pace of life: faster vs. slower) × 2 (Temporal focus: past vs. future) two-way ANOVA showed significant two-factor interaction, *F* (1, 270) = 55.11, *p* = 0.000, $\eta_{p}^{2}$ = 0.09. Further simple effects analysis showed that when participants were fasterlife -paced, the future temporal focus group (*M* = −4.43, *SD* = 1.83) was more likely than the past temporal focus group (*M* = −6.00, *SD* = 1.29) to prefer larger gains in the distant future, *t* (1, 270) = 46.81, *p* = 0.000. However, when participants were living at a slower pace, the difference between the future time focus group (*M* = −5.69, *SD* = 1.51) and the past time focus group (*M* = −5.65, *SD* = 1.23) was not significant, *t* (1, 270) = 0.05, *p* = 0.83. Results are shown in Table 4 and Figure 3.

The results of Study 3 confirmed that the pace of life can influence intertemporal decision making. People with a faster life pace, relative to those with a slower life pace, prefer the SS option in an intertemporal choice. Temporal focus can influence intertemporal decision making, and individuals with a future focus prefer SS to those with a past focus. The pace of life and temporal focus can jointly influence individuals’ intertemporal decision-making preferences. However, individuals with a slow-pace-of-life strategy were less influenced by temporal focus and did not differ significantly in their intertemporal decision preferences regardless of the temporal focus. This result partially confirmed hypothesis 3.

## 5. Discussion

### 5.1. The Impact of the Pace of Life on Intertemporal Decision Making

Our main result confirmed that time scarcity can cause individuals to have less control and thus a preference for SS [7] and the faster the pace of life, the more time is scarce. This study examined the effect of the pace of life on intertemporal decision making. Study 1 used a correlational study to find that the faster the pace of life, the more individuals preferred SS, and the slower the pace of life, the more they preferred LL. The results of Experiment 2 and Experiment 3 also support this phenomenon. There might be four possible reasons, as follows:

First of all, fast-paced individuals need to deal with more things per unit of time, and time is even more scarce. The opposite is true for a slow pace of life. Research has shown that time deprivation takes away attentional resources and leads to the phenomenon of “participative time stretching” [26]. The perceived time-based model (PTBM) argues that participative time extension will increase individuals’ impulsivity and make them unwilling to wait [27,28] and thus opt for immediate gains [29]. Then, for fast-paced individuals, the perception of longer time periods makes them unwilling to wait, and therefore they prefer SS. Furthermore, the lack of temporal resources may cause individuals to pay more attention to the temporal dimension in intertemporal decision making. The single dimension priority model suggests that in intertemporal decision making, the decision maker needs to compare the options in the delay dimension and the outcome dimension, and then make the choice based on the dominant dimension [30]. Then, individuals with a fast pace of life are overly concerned about time, which may lead them to make decisions based on the time dimension and therefore prefer immediate rewards. However, individuals with a slow pace of life have sufficient time resources and thus may be more concerned with the outcome dimension and prefer LL in intertemporal decision making. Furthermore, life history strategy theory holds that greater environmental stressors can be impulsive and also make individuals prioritize short-term preferences [31,32]. Fast-paced individuals are more impulsive and prioritize short-term preferences due to their higher environmental stress, which makes them more attracted to SS. Finally, it has been found that a fast pace of life brings about greater time pressure [11]. High time pressure will trigger anxiety [33]. Individuals may choose to act immediately to relieve anxiety and therefore prefer immediate rewards. In addition, high temporal pressure will also prompt individuals to be more impulsive in intertemporal choices, thus choosing immediate rewards rather than waiting [7,34,35].

### 5.2. Manipulations in Time Perception Styles Have Different Effects on Individuals with Different Paces of Life

It was found that by manipulating the way individuals perceive time, we were able to influence participants’ intertemporal decision preferences. This study separately intervened in the view of time or temporal focus of individuals with different paces of life. The results of Study 2 found that the time perspective and pace of life can jointly influence intertemporal decision preferences. To be specific, fast-paced individuals preferred small near-term rewards under the manipulation of a linear time perspective and large far-term rewards under the manipulation of a circular time perspective. To our surprise, there was no change in intertemporal decision preferences with slow-paced individuals after either the linear or circular time perspective manipulations. The results of Study 3 also found that the temporal focus and pace of life could jointly influence intertemporal decision preferences. Specifically, fast-paced individuals preferred small near-term rewards with the future time-focus manipulation and large far-term rewards with the past time-focus manipulation. There was no change in intertemporal decision preferences with slow-paced individuals after receiving either future or past time focus manipulations.

The reasons for this are as follows. For fast-paced individuals, they reported feeling greater time pressure [11] and a need to deal with more tasks in a short period of time. Therefore, they may need to weigh various factors to find the best strategy. The above characteristics make them more easily influenced and more flexible in decision making when receiving manipulations. In addition, as previously mentioned, the fast pace of life will lead to more time scarcity, which in turn takes up cognitive resources so that they only focus on the present and have less control [7]. Then, this also makes them pay more attention to the outcome of the manipulation in the present, unable to adhere to their preferences, and prone to uncontrolled changes in decision preferences due to various factors. However, for slow-paced individuals, living in a time-sufficient environment, there is more control. Therefore, even with successful manipulations of their time perspective or time focus, preferences for long-term intertemporal decision making are not easily changed. In addition, individuals with a slow pace of life have sufficient time, which makes them less concerned about time, and based on the unidimensional dominance model, they may be more concerned about the outcome of the decision and therefore prefer LL. Then, even after successful manipulations on their temporal view or temporal focus, the concern about the outcome of the decision fails to change, which leaves their intertemporal decision preferences unchanged.

### 5.3. Implication and Limitations

With the accelerated pace of life, people are increasingly impatient in the face of decision making. This study is important for how to improve people’s decision-making preferences and guide them to make reasonable decisions. 

The results of the study suggest that the accelerated pace of life has a significant impact on individuals’ intertemporal decision making. This prompts us to be aware of the negative effects of a faster pace of life, such as more difficulty in delaying gratification. In addition, we improved individuals’ intertemporal decision making by changing the way they perceive time (time perspective and time focus). In daily life, individuals with different pace-of-life rhythms can improve their decision preferences by means of manipulations, and thus make more reasonable and correct decisions. This also provides an idea for scholars to explore more ways to guide people to make rational decisions. 

The present study also has some limitations. First, research has shown that the research paradigm of intertemporal decision making suffers from the problem of consistent utility between real and hypothetical situations, with differences between hypothetical and real discount scenarios [36,37]. This suggests that the ecological validity of the present study may be problematic and that future studies could further examine it in the context of real delayed discounting situations. Secondly, the pace of life may change with the environment and other factors. In the future, it could be explored whether individuals change their intertemporal decision preferences in response to changes in their pace of life to establish a stable link between pace of life and intertemporal decision making. Another limitation is the low Cronbach α of the Pace of Life Questionnaire used in our study. Future studies will revise this scale and verify our questions again. Finally, this study found that temporal cognitive factors such as views of time or temporal focus can change individuals’ intertemporal decision preferences, but the rationale for this has not been examined. In the future, we can further examine whether individual participative time perception differences or other variables play a key role in this and provide more theoretical basis for manipulating time perception factors to change intertemporal decision preferences.

## 6. Conclusions

The pace of life can influence intertemporal decision preferences, with faster-paced individuals preferring small near-term rewards and slower-paced individuals preferring large far-term rewards. The view of time or temporal focus manipulation changed the intertemporal decision preferences of faster-paced individuals to prefer small near-term rewards under a linear view of time or future temporal focus and large distant rewards under a circular view of time or past temporal focus. However, slower-paced individuals did not change their intertemporal decision preferences after receiving the temporal perspective or the temporal focus manipulation. The entire study identified the role of the pace of life on intertemporal decision making from the perspective of resource scarcity and the different effects of views of time and temporal focus manipulation on intertemporal decision making of individuals with different paces of life from the perspective of differences in people’s perception of time itself.

## Figures and Tables

**Figure 1 ijerph-20-04301-f001:**
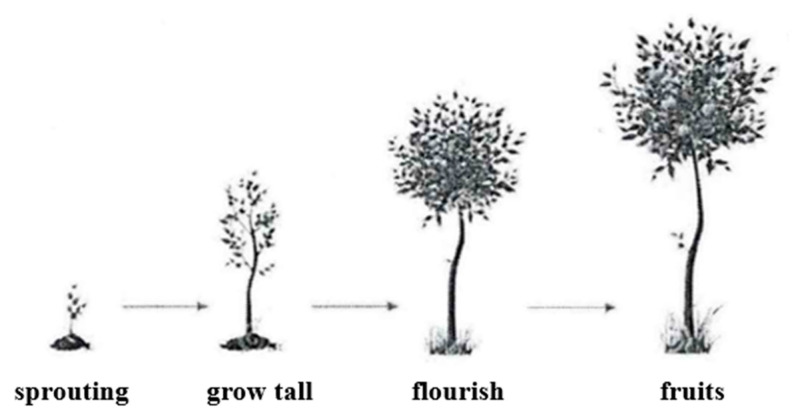
Circular time view manipulation.

**Figure 2 ijerph-20-04301-f002:**
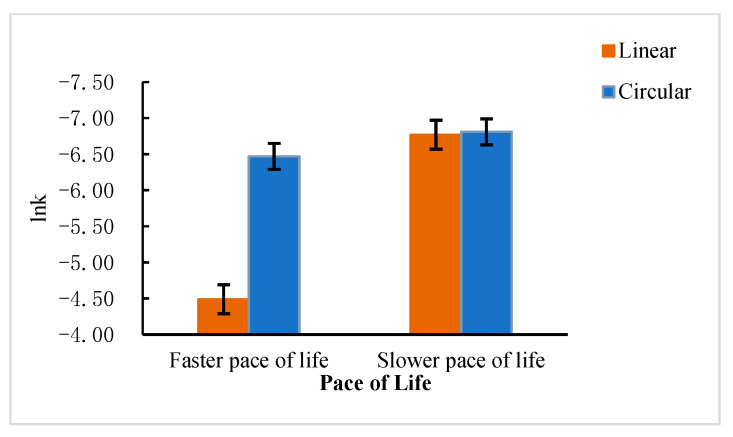
Comparison of participants’ intertemporal decision scores (lnK) under different paces of life and different views of times.

**Figure 3 ijerph-20-04301-f003:**
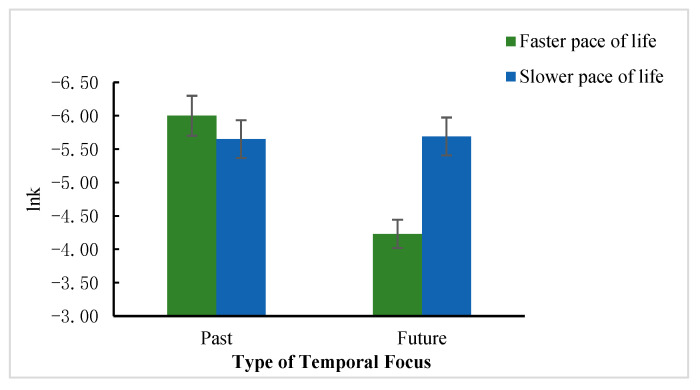
Comparison of participants’ intertemporal decision scores (lnK) under different paces of life and different temporal focuses.

**Table 1 ijerph-20-04301-t001:** Participant’s grouping information (*n* = 303).

	Grouping			
	Slower-Linear	Slower-Circular	Faster-Linear	Faster-Circular
*n*	76	76	74	77
%	25.1%	25.1%	24.4%	25.4%

**Table 2 ijerph-20-04301-t002:** Intertemporal decision scores (lnK) for participants with different paces of life under different views of time conditions (*M* ± *SD*).

Type of View of Time	Faster Pace of Life	Slower Pace of Life
Linear	−4.49 ± 1.74	−6.77 ± 1.46
Circular	−6.47 ± 1.36	−6.81 ± 1.35

**Table 3 ijerph-20-04301-t003:** Participant’s grouping information (*n* = 274).

	Grouping			
	Slower-Future	Slower-Past	Faster-Future	Faster-Past
*n*	76	76	74	77
%	27.7%	27.7%	27.0%	28.1%

**Table 4 ijerph-20-04301-t004:** Intertemporal decision scores (lnK) for participants with fast and slow paces of life in different temporal focus conditions (*M* ± *SD*).

Type of Temporal Focus	Faster Pace of Life	Slower Pace of Life
Past	−6.00 ± 1.29	−5.65 ± 1.23
Future	−4.23 ± 1.83	−5.69 ± 1.51

## Data Availability

The original contributions presented in the study are included in the article, and further inquiries can be directed to the corresponding author/s.

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
