# Peer review of "View of Times and Temporal Focus under the Pace of Life on the Impact of Intertemporal Decision Making"

_ijerph, 2023, doi:10.3390/ijerph20054301_

Round 1

Reviewer 1 Report

This is an interesting topic. There are studies looking at life history and temporal discounting, e.g. doi: 10.3390/bs12030063 (very recent) and 
Conceptualizing time preference: a life-history analysis. Copping LT, Campbell A, Muncer S. Evol Psychol. 2014 Sep 29;12(4):829-47. The authors should link their pace of life with the life history work.

Regarding study 2, your results show a trait (pace of life) by state (linear or circular intervention) interact only for the fast pace of life folks, i.e. those with a fast pace of life can achieve a similar discounting score as the slow pace of life folks if they are presented with circular time. That sounds like an efficient manipulation.

This is similar for study 3, i.e. fast pace of life group but not slow pace of life group (a trait) are affected by a state manipulation: past vs future.

method section: please provide the software used through which you administered the survey. Were students tested online, individually or at campus? How long did it take to fill out the survey? What software did you use to analyse the data?

Please add as limitation that you used only a questionnaire / hypothethical discounting, e.g., Lagorio CH, Madden GJ. Delay discounting of real and hypothetical rewards III: steady-state assessments, forced-choice trials, and all real rewards. Behav Process. 2005;69(2):173–87 and Matusiewicz AK, et al. Statistical equivalence and test-retest reliability of delay and probability discounting using real and hypothetical rewards. Behav Process. 2013;100:116–22.

Another limitation is the .6 internal consistency of the pace of life scale. Did you check whether it improves if you leave one item out? Since you used this scale in all three studies, it is a bit worrisome that it rests on such an unreliable scale. If internal consistency improves after leaving one item out, please use the abbreviated version in all your analysis.

Regarding study 3, you split by the mean, but you should split by the median, not mean (doubt it changes the story), or use it as a co-variate, you can keep the plots (for plotting purposes we split into two groups...) but for the analysis you are not using a 2x2 design but an ANCOVA (pace of life scale as covariate). This also applies for study 2.

minor
line 82 "the more he may ..." should be "the more they may ..."

line 105-109: please provide the female:male info after 799 (final sample size), not after 991 (as the numbers for female and male make up 799 but not 991)

line 159: p < .001, please also report Cohen's d (effect size), same for line 319

line 215: p < .001, please report eta2 (or partial eta square)

Reviewer 2 Report

Thanks for your efforts on preparation of this nice paper. There are some points which need revisions before publication.

- In the first sentence of abstract "e xplored" must be written as "explored".

- There must be connection sentence between the second and third sentences of the abstract. Study 1, 2 and 3 seems independent of the beginning part in the current form.

- In the sentence in the first paragraph of introduction starting with "previous studies ..." you should place citations in more proper places i.e. with their relations to the effective factors.

- Who says “Time is money and efficiency is life”? Place a proper citation.

- There are some errors with punctuations. Have a complete look at the manuscript within this context.

- You must mention the number of studies you conduct within this research in a proper place in the second paragraph of the introduction.

- Presentation of hypotheses can be given in a better shape.

- Introduction part must provide the contributions of the study and it should be ended with an organization paragraph.

- The beginning paragraph of section 2 can contain some more basic information about Study 1.

- There is problem with the number of the subtitle for results of study 1. Please check it.

- Similarly, the beginning paragraph of section 3 can contain some more basic information about Study 2.

- Similarly, the beginning paragraph of section 4 can contain some more basic information about Study 3.

- The numerical information about participants of each study can be presented better with tables.

- The language quality of the manuscript can be improved by the help of a native speaker. There are some sentences which need improvement of language quality like "The reasons for this are as follows..." in line 405. I recommend a general check for the paper.

Reviewer 3 Report

Recommendation: Major Revision

 Comments:

1.        Figure 2 and 3 are difficult to read. Please distinguish the lines by color.

2.        Please provide some direction for future research.

3.        Please cite more recent references from The International Journal of Environmental Research and Public Health to show the relevance of your study for the journal.

4.        Please fill in the complete information : Author ContributionsFundingInstitutional Review Board StatementInformed Consent StatementData Availability StatementConflicts of Interest.

Round 2

Reviewer 3 Report

Recommendation : Accept